# Causal Motion Tokenizer for Streaming Motion Generation

## Abstract

Recent advancements in human motion generation have leveraged various multimodal inputs, including text, music, and audio. Despite significant progress, the challenge of generating human motion in a streaming context—particularly from text—remains underexplored. Traditional methods often rely on temporal modalities, leaving text-based motion generation with limited capabilities, especially regarding seamless transitions and low latency. In this work, we introduce MotionStream, a pioneering motion-streaming pipeline designed to continuously generate human motion sequences that adhere to the semantic constraints of input text. Our approach utilizes a Causal Motion Tokenizer, built on residual vector quantized variational autoencoder (RVQ-VAE) with causal convolution, to enhance long sequence handling and ensure smooth transitions between motion segments. Furthermore, we employ a Masked Transformer and Residual Transformer to generate motion tokens efficiently. Extensive experiments validate that MotionStream not only achieves state-of-the-art performance in motion composition but also maintains real-time generation capabilities with significantly reduced latency. We highlight the versatility of MotionStream through a story-to-motion application, demonstrating its potential for robotic control, animation, and gaming.

## 1 Introduction

Recent progress in AI, driven by large-scale models OpenAI (2023); Touvron et al. (2023a;b), has given network models initial intelligent "thoughts" Wei et al. (2022), offering hope for developing world models and foundation models Ha & Schmidhuber (2018); Majumdar et al. (2024), which has sparked interest in studying humanoid robotics Darvish et al. (2023); Zhang et al. (2023a); Mu et al. (2024). As one method for controlling humanoid agents, human motion generation has made significant advancements, enabling the creation of human motion under various conditions such as text Zhang et al. (2023b); Guo et al. (2023), music Gong et al. (2023); Zhou & Wang (2023), audio Yi et al. (2023); Yin et al. (2023), and motion Liu et al. (2024); Chen et al. (2023a). Given one steaming modality, this human motion model, capable of generating in a streaming fashion, should benefit both virtual humanoid agents and humanoid robotics in terms of behavioral outputs.

Previous motion researches focus on various tasks such as single-clip motion generation from actions Petrovich et al. (2021b); Guo et al. (2020); Athanasiou et al. (2022a); Xin et al. (2023); Lee et al. (2023); Wang et al. (2022a) or text Guo et al. (2022a); Zhang et al. (2022); Tevet et al. (2022); Petrovich et al. (2022); Lu et al. (2023); Guo et al. (2023), motion composition Athanasiou et al. (2022a); Shafir et al. (2023b); Barquero et al. (2024), motion prediction Zhang et al. (2021); Chen et al. (2023a), and multi-track motion generation Petrovich et al. (2024). Some motion composition studies Athanasiou et al. (2022a); Lee et al. (2023); Qian et al. (2023); Li et al. (2023) focused on explicitly modeling subsequent transition and motion by current motions. However, they require datasets with multiple consecutive annotated motions, making it challenging to achieve smooth transitions. For example, TEACH applies interpolation techniques like Slerp to mitigate misalignment between motion segments. Other methods generate complete motions under multiple conditions by interpolating or stitching together motions generated from a single condition. DoubleTake Shafir et al. (2023b) utilizes a diffusion-based motion generator (MDM Tevet et al. (2022)) to create motion clips and further combine them with diffusion-denoising. However, this framework, neither causal nor steaming generation, affects both generated and current motions. FlowMDM Barquero et al. (2024) introduces a temporal attention mechanism to ensure each frame aligns with texts for

better motion-text alignment and smoother transitions. However, it processes all conditions simultaneously, resulting in longer generation latency as the number of conditions increases. To address the above limitations, we focus on developing a streaming motion generator capable of progressively generating human motion sequences with low latency based on text descriptions.

Our motivation stems from translating a lengthy textual narrative like Story-to-Motion Qing et al. (2023), detailing a series of human activities into seamless, lifelike human motions that hold potential for robotic control, virtual animation, and gaming. However, achieving this requires overcoming two critical challenges. The first is ensuring smooth transitions between each motion segment while accurately reflecting the corresponding text conditions. The second is maintaining low and consistent generation latency, even as the number of text instructions increases.

In this work, we introduce MotionStream, a motion-streaming pipeline designed to generate naturally continuous motions that faithfully adhere to the semantic constraints of continuous text input. To output a motion clip seamlessly with the adjoining motions, we first develop a causal motion tokenizer to construct our causal motion codebook. More specifically, our tokenizer is built upon residual vector quantized variational autoencoder (RVQ-VAE). We further develop a dual transformer scheme to accurately predict causal motion tokens from the given textual inputs, effectively translating complex textual descriptions into corresponding dynamic motions. This dual approach not only enriches the motion quality but also maintains semantic fidelity across the generated motion sequences. The motion tokenizer employs causal convolution, greater code distance, and a replacing scheme during training to enhance the handling of long sequences. Additionally, to further improve transition smoothness, we incorporate memory tokens during mask modeling. Then, for motion generation under semantic text conditions, we adopt a BERT-like Masked Transformer and a Residual Transformer following Momask Guo et al. (2023), , which are specialized in generating motion tokens for the base VQ layer and the residual layers, respectively. Extensive experiments demonstrate that MotionStream not only achieves state-of-the-art performance in motion composition but also maintains high generation efficiency and effectiveness.

We summarize our contributions as follows: (1) We introduce MotionStream, a new casual and steaming motion generator that continuously produces motion of arbitrary length, without relying on explicit labeling on transitions between motions. (2) We design our Causal Motion Tokenizer for long motion decoding, which improves the transition smoothness of streaming motion outputs. (3) Our extensive evaluation shows that MotionSteam outperforms diffusion-based models in efficiency, supports real-time motion streaming with ∼0.2s generation latency, and achieves state-of-the-art performance on the BABEL and HumanML3D datasets. We showcase a story-to-motion application driven by instructions from GPT-4 to demonstrate the versatility of MotionSteam.

## 2 RELATED WORK

### 2.1 HUMAN MOTION SYNTHESIS

Motion generation from multi-modal inputs such as text Petrovich et al. (2023); Jiang et al. (2023); Chen et al. (2023b), speech Chen et al. (2024); Yi et al. (2023), music Aristidou et al. (2022), images Jiang et al. (2024), and videos Mehta et al. (2020) entails synthesizing dynamic human activities by leveraging diverse data types, which considerably enhances the applicability and realism of the generated movements. Predicated on distinct classification paradigms, this process can be delineated as either conditional Guo et al. (2020); Wang et al. (2022b) or unconditional Urtasun et al. (2007); Shi et al. (2020), unimodal Petrovich et al. (2023); Chen et al. (2023b) or multimodal Kritsis et al. (2021); Wu et al. (2024), and involves static Jiang et al. (2024) or dynamic Mehta et al. (2020) input, collectively underscoring the adaptability of the motion synthesis mechanisms and enabling the creation of contextually responsive and data-informed human movements.

Among the various paradigms for motion generation, utilizing textual inputs is notably prevalent due to their capacity to richly describe complex human behaviors and emotional states. While advanced models such as GANs Xu et al. (2023); Barsoum et al. (2018), VAEs Petrovich et al. (2021a); Bie et al. (2022), and diffusion Zhang et al. (2022); Xin et al. (2023) methods effectively translate textual narratives into dynamic motions, traditional methods still encounter significant challenges with seamless transitions and maintaining low latency, especially in real-time streaming contexts.

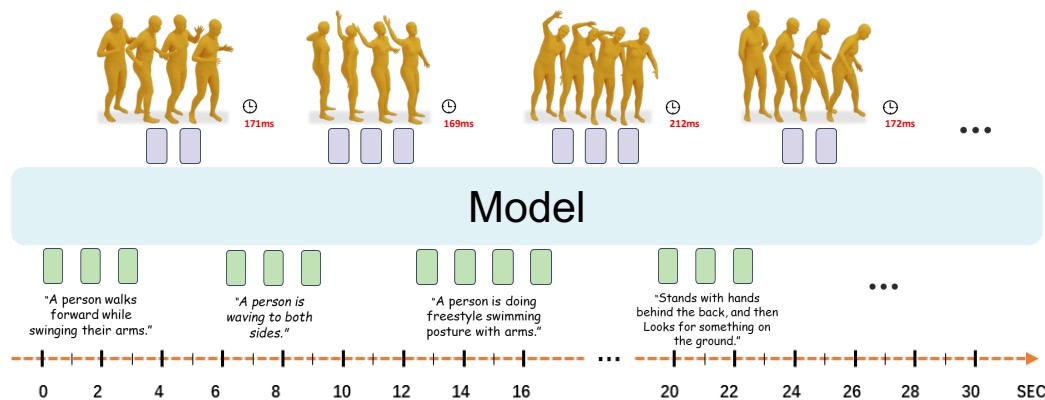

Figure 1: Method overview: MotionStream consists of a motion tokenizer $\mathcal{V}$ (Section 3.1) a Mask Transformer (Section 3.2) and a Residual Transformer (Section 3.3). MotionStream is capable of producing seamless and dynamic motions driven by narrative descriptions.

## 2.2 MOTION COMPOSITION

Motion composition involves synthesizing coherent sequences from discrete motion segments, a process complicated by the scarcity of suitable training data. This synthesis often requires integrating motions conditioned on both actions and textual descriptions.

Diffusion models like EDGE Tseng et al. (2023) and PriorMDM Shafir et al. (2023a) are prominent for their ability to ensure smooth transitions by enforcing temporal constraints at the junctions of motion segments. These models excel at creating fluid motion sequences from extensive textual inputs by blending multiple motion clips over time Zhang et al. (2023d); Rombach et al. (2022). However, they are less effective in environments requiring adaptation to real-time, continuous text streams, due to their inherent design which primarily handles pre-segmented input scenarios.

Auto-regressive methods, such as those employed by TEACH Athanasiou et al. (2022b) and EMS Qian et al. (2023), sequentially generate motions, with each segment conditioned on its predecessor. TEACH generates one motion at a time per text prompt, while EMS utilizes a two-stage approach to first generate and then merge actions, which aids in maintaining coherence across the sequence. Despite their precision in controlled environments, these models struggle with real-time responsiveness, as they rely on processing a series of predetermined inputs rather than adapting on-the-fly to incoming data streams.

Both diffusion and auto-regressive methods, while capable, primarily compose motion by stitching multiple segments across different times or generating complex motions from lengthy texts simultaneously. This technique limits their adaptability and responsiveness, particularly in dynamic environments where continuous and real-time text input integration is crucial.

## 2.3 VECTOR QUANTIZATION

Vector Quantization (VQ) Gray (1984); Esser et al. (2021) simplifies data by mapping vectors to fewer representative centroids, and extending this, Residual Vector Quantization (RVQ) Barnes et al. (1996); Lee et al. (2022) enhances precision by encoding the residuals. This advanced approach underpins our use of sophisticated encoding strategies for motion synthesis. In the TM2T Guo et al. (2022c) project, a Vector Quantized Variational Autoencoder (VQ-VAE) Van Den Oord et al. (2017a) accurately maps human motions to discrete tokens, improving codebook selection. T2M-GPT Zhang et al. (2023b) further refines this by incorporating Exponential Moving Average (EMA) Nakano et al. (2017) and code reset techniques to reduce quantization errors and enhance reconstruction fidelity.

Building on these innovations, the integration of memory tokens with a BERT-like Devlin (2018) Masked Transformer and a Residual Transformer enables precise motion generation from semantic text inputs, significantly advancing our capabilities in high-precision motion composition.

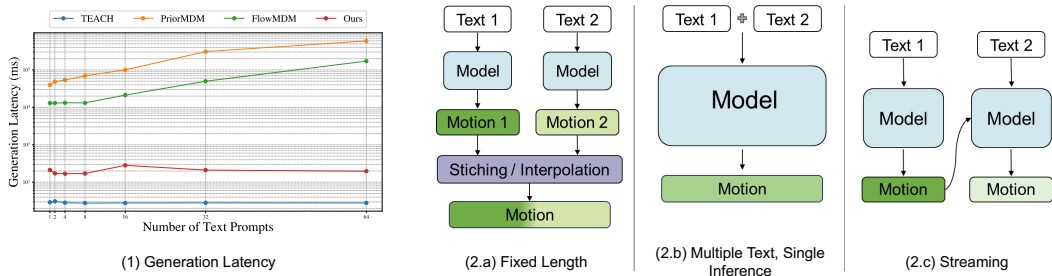

Figure 2: Motion Generation Approaches and Latency Performance Overview. (1) Generation latency versus number of text prompts on a V100 machine. (2.a) Generation of individual motions from separate text prompts, combined via stitching or interpolation. (2.b) Approach processing multiple texts in a single inference step to generate a whole motion sequence. (2.c) Approach to generate continuous motion from consecutive text inputs without post-processing stitching.

## 3   METHOD

We introduce MotionStream, a real-time motion generation framework to synthesize human motion sequences in a streaming format. As depicted in Fig. 4, MotionStream incorporates a causal motion Vector Quantized Variational Autoencoder (VQ-VAE) (Section 3.1) that encodes raw motion data into multi-layered causal motion tokens and reconstructs these tokens into continuous motion sequences. A masked transformer (Section 3.2) is utilized to generate base layer motion tokens, while a residual transformer (Section 3.3) processes tokens for the subsequent layers, ensuring seamless motion generation under text conditions.

The causal motion tokenizer within MotionStream consists of an encoder, $\mathcal{E}_m$, and a decoder, $\mathcal{D}_m$. The encoder $\mathcal{E}_m$ transforms $L$ frames of raw motion, $m^{1:L} = \{x^i\}_{i=1}^L$, into $L$ latent vectors, which are quantized into discrete motion tokens, $z^{1:L}$, utilizing a learnable codebook $Z = \{z^i\}_{i=1}^K \subset \mathbb{R}^d$. The decoder $\mathcal{D}_m$ then reconstructs the motion sequence $\hat{m}^{1:L} = \mathcal{D}(z^{1:L})$, preserving temporal coherence and physical plausibility. Given $S$ text sentences, $w_s^{1:N_s}$, each with a length $N_s$ describing text instructions for motion segments, MotionStream aims to generate $S$ segments of motion tokens, $\hat{x} = \{\hat{x}_1, \hat{x}_2, \ldots \hat{x}_S\}$, which can subsequently be decoded into motion segments, $\hat{m} = \{\hat{m}_1, \hat{m}_2, \ldots \hat{m}_S\}$, corresponding to each text instruction. These segments are expected to exhibit plausible and smooth transitions between each motion segment, ensuring a cohesive and fluid overall motion sequence.

### 3.1   CAUSAL MOTION TOKENIZER

To represent motion in discrete tokens, we pre-train a 3D human motion tokenizer $\mathcal{V}$ utilizing a Residual Vector Quantization (RVQ) framework, building on the VQ-VAE architecture as introduced in Van Den Oord et al. (2017b); Siyao et al. (2022); Guo et al. (2022b); Zhang et al. (2023b); Guo et al. (2023). This tokenizer is composed of an encoder $\mathcal{E}_m$, a residual vector quantizer, and a decoder $\mathcal{D}_m$, all tailored for optimal performance in learning causal motion tokens.

The encoder $\mathcal{E}_m$ processes raw motion sequences into latent representations by applying 1D causal convolutions along the temporal dimension of the input motion features $m^{1:M}$. These causal convolutions capture the temporal dependencies between consecutive frames, ensuring that each frame is influenced only by preceding frames. This is crucial for preserving the causal structure required for streaming motion generation tasks. Once the latent vectors $\hat{z}^{1:L}$ are derived from the encoder, Residual Vector Quantization (RVQ) is employed. Unlike conventional vector quantization, RVQ decomposes the latent representation across multiple stages of quantization, allowing for progressive refinement of the representation through residual encoding. The quantization procedure utilizes a learnable codebook $Z = \{z^i\}_{i=1}^K \subset \mathbb{R}^d$, where $K$ denotes the number of discrete codebook entries and $d$ represents the dimensionality of each embedding. Additionally, the high-dimensional latent vectors are downsampled to a lower-dimensional latent space before quantization, enhancing the efficiency of feature learning for causal motion tokens, as detailed in the supplementary materials. The quantization function $Q(\cdot)$ iteratively maps each latent vector $\hat{z}^i$ to its nearest codebook

Figure 3: The architecture of MotionStream's motion tokenizer, $\mathcal{V}$, detailed in Section 3.1. It showcases the Residual Vector Quantization (RVQ) framework employed by the tokenizer, which includes both an encoder and a decoder equipped with causal convolutions. This design enables the effective encoding and decoding of motion data, ensuring temporal coherence and continuity in the generated motion sequences.

entry $z_k \in Z$, progressively refining the representation through residual stages. This process is mathematically expressed as:

$$z_i = Q(\hat{z}^i) := \arg \min_{z_k \in Z} \|\hat{z}_i - z_k\|_2 . \tag{1}$$

The decoder $\mathcal{D}_m$ reconstructs the motion sequence $\hat{m}^{1:M} = \mathcal{D}_m(z^{1:L})$ from the quantized latent vectors $z^{1:L}$. Similar to the encoder, the decoder applies causal convolutions to ensure the preservation of temporal dependencies during the reconstruction process, thereby maintaining the causal integrity of the motion sequence. This causal structure is essential for real-time motion generation tasks. To train our proposed motion tokenizer, we introduce a novel training paradigm that enhances the quality and diversity of the generated motion sequences by optimizing three key loss components: reconstruction loss $\mathcal{L}_r$, embedding loss $\mathcal{L}_e$, and commitment loss $\mathcal{L}_c$. The overall loss function is defined as $\mathcal{L}_{\mathcal{V}} = \mathcal{L}_r + \mathcal{L}_e + \mathcal{L}_c$.

**Code Masking.** In addition to implementing quantization layer dropout as described in Guo et al. (2023), we introduce a layer-specific code masking strategy. This strategy is motivated by the goal of enhancing the model's ability to reconstruct and infer from incomplete data, thereby learning more robust and essential features of the motion. Consequently, during training, certain portions of the motion codes are masked and substituted with randomly selected tokens from the same codebook. The decoder is then tasked with reconstructing the entire motion sequence, accommodating these modifications to enhance its robustness and ability to handle noisy input effectively. **Double Round Training.** Our training process also introduces a unique methodology for causal motion tokenizer, ensuring that the model learns to generate smooth and continuous motion across variable segments. We first randomly split the input motion sequence into two subsequences, After splitting, we conduct two forward passes to process the resulting motion segments separately. First, the initial part of the sequence $m_{\text{head}}$ is passed through the motion VAE followed by resetting the causal convolution in Encoder leaving Decoder alone for a continuous generation. The second part of the sequence $m_{\text{tail}}$ is then processed in a similar manner. By splitting and processing the motion sequences in this manner, we introduce variability in the sequence lengths and transitions, improving the model's ability to generate high-quality, temporally coherent motion for real-time applications.

## 3.2 MASK TRANSFORMER

Utilizing this motion tokenizer, we transform human motion sequences $m^{1:M}$ into sequences of motion tokens $z^{1:L}$. These tokens are represented as layers of sequences of indices, where each index corresponds to a specific motion token within a layer. In alignment with Guo et al. (2023), our approach models the base-layer motion tokens $x_0^{1:L}$ using a masked transformer. During preprocessing, we randomly replace some tokens with a special [MASK] token to facilitate learning. The masking ratio is adjusted dynamically using a cosine function, $\gamma(\tau)$, where $\tau$ is sampled from a uniform distribution $U(0, 1)$, allowing for variable sequence corruption. The training strategy in-

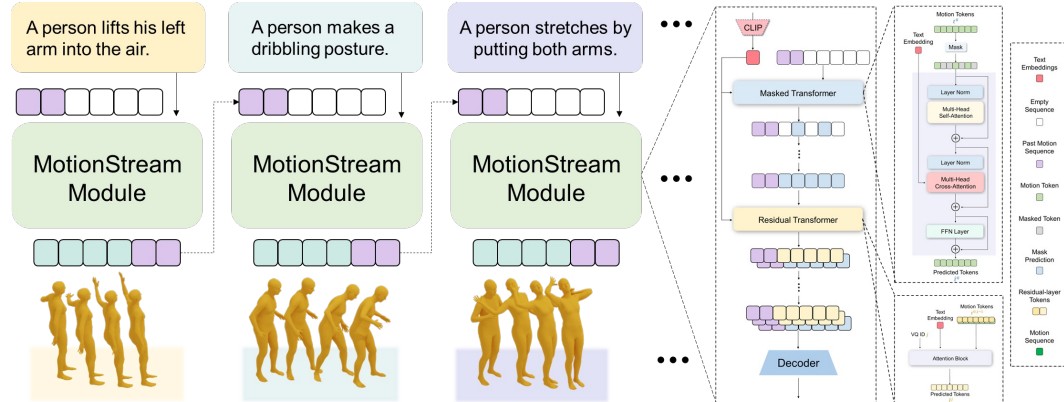

Figure 4: Method overview: In addition to the motion tokenizer, a dual transformer scheme is proposed to accurately predict causal motion tokens from the given textual inputs, effectively translating complex textual descriptions into corresponding dynamic motions.

cludes masking 80% of the selected tokens, replacing 10% with random tokens, and leaving 10% unchanged. Post-masking, the objective is to predict the masked tokens based on the associated text $w_s$ and the modified sequence $\tilde{x}^0$. Text features are extracted using CLIP Radford et al. (2021), and the transformer is optimized to minimize the negative log-likelihood of the predictions according to:

$$\mathcal{L}_{\text{mask}} = \sum_{\tilde{x}_k^0 = [\text{MASK}]} - \log p_\theta(x_k^0 | \tilde{x}^0, w_s). \tag{2}$$

**Tokens Compression.** Our motivation stems from addressing the substantial temporal redundancy observed in human motion sequences. Directly downsampling these sequences before processing significantly compromises the reconstruction capabilities of our causal motion tokenizer, as elaborated in the Appendix. This reduction in performance arises because each causal token is required to encapsulate significant information from both the current and preceding frames to maintain distinctiveness within the motion codebook. However, such detailed information is unnecessary during the mask modeling stage, where the focus is on mapping text to tokens rather than capturing motion nuances. Therefore, we optimize the process by downsampling the causal tokens before they enter the mask transformer and subsequently upsampling them to preserve essential temporal details.

**Condition Injection.** Prior research Guo et al. (2023) has employed a method of incorporating text conditions into Mask Transformer by concatenating the pooled features from the CLIP text encoder with the masked token features. This method, however, has limitations in retaining complete text information. To address this, we introduce a more effective condition injection technique that enhances text retention and is particularly suitable for complex text instructions. We first process the motion token features through the self-attention mechanism. Subsequently, these features undergo cross-attention with the last hidden layer of the CLIP text encoder, thereby preserving a richer textual context. This method ensures that more comprehensive text details are integrated into the motion tokens, potentially improving the fidelity and relevance of the generated motion sequences.

**Memory Tokens.** During inference, we start with an initially masked sequence $x^0(0)$ and aim to construct the base-layer token sequence $x^0$ over $M$ iterations. The Mask Transformer calculates the probability distribution of tokens at masked locations, selectively sampling and re-masking tokens based on confidence levels. This process is repeated, using the updated token sequence $x^0(l+1)$ for subsequent predictions until completion after $M$ iterations. For subsequent text prompts, the final frames from previously generated motion sequences are utilized as memory tokens to inform the generation of new tokens. These memory tokens, in conjunction with newly masked tokens, facilitate the prediction of the next set of motion tokens.

| | Subsequence | | | | Transition | | | |
|---|---|---|---|---|---|---|---|---|
| | R-prec ↑ | FID ↓ | Div → | MM-Dist ↓ | FID ↓ | Div → | PJ → | AUJ ↓ |
| GT | $0.796^{\pm0.004}$ | $0.00^{\pm0.00}$ | $9.34^{\pm0.08}$ | $2.97^{\pm0.01}$ | $0.00^{\pm0.00}$ | $9.54^{\pm0.15}$ | $0.04^{\pm0.00}$ | $0.07^{\pm0.00}$ |
| DoubleTake* | $0.643^{\pm0.005}$ | $0.80^{\pm0.02}$ | $9.20^{\pm0.11}$ | $3.92^{\pm0.01}$ | $\underline{1.71}^{\pm0.05}$ | $\mathbf{8.82}^{\pm0.13}$ | $0.52^{\pm0.01}$ | $2.10^{\pm0.03}$ |
| DoubleTake | $0.628^{\pm0.005}$ | $1.25^{\pm0.04}$ | $9.09^{\pm0.12}$ | $4.01^{\pm0.01}$ | $4.19^{\pm0.09}$ | $8.45^{\pm0.09}$ | $0.48^{\pm0.00}$ | $1.83^{\pm0.02}$ |
| MultiDiffusion | $0.629^{\pm0.002}$ | $1.19^{\pm0.03}$ | $\mathbf{9.38}^{\pm0.08}$ | $4.02^{\pm0.01}$ | $4.31^{\pm0.06}$ | $8.37^{\pm0.10}$ | $0.17^{\pm0.00}$ | $1.06^{\pm0.01}$ |
| DiffCollage | $0.615^{\pm0.005}$ | $1.56^{\pm0.04}$ | $8.79^{\pm0.08}$ | $4.13^{\pm0.02}$ | $4.59^{\pm0.10}$ | $8.22^{\pm0.11}$ | $0.26^{\pm0.00}$ | $2.85^{\pm0.09}$ |
| FlowMDM | $\underline{0.685}^{\pm0.004}$ | $\underline{0.29}^{\pm0.01}$ | $9.58^{\pm0.12}$ | $\underline{3.61}^{\pm0.01}$ | $\mathbf{1.38}^{\pm0.05}$ | $\underline{8.79}^{\pm0.09}$ | $\underline{0.06}^{\pm0.00}$ | $\underline{0.51}^{\pm0.01}$ |
| MotionStream | $\mathbf{0.719}^{\pm0.005}$ | $\mathbf{0.13}^{\pm0.02}$ | $\underline{9.27}^{\pm0.11}$ | $\mathbf{3.36}^{\pm0.01}$ | $2.56^{\pm0.05}$ | $7.93^{\pm0.05}$ | $\mathbf{0.05}^{\pm0.01}$ | $\mathbf{0.38}^{\pm0.03}$ |

Table 1: Comparison of motion composition on HumanML3D Guo et al. (2022a) dataset. The arrows (→) indicate that closer to *Real* is desirable. **Bold** and underline indicate the best and the second best result on text-to-motion task.

### 3.3 RESIDUAL TRANSFORMER

Following the extraction of base layer motion tokens using the Mask Transformer, we implement a residual transformer to process tokens across multiple residual quantization layers, each tailored to capture varying levels of motion complexity. This setup, detailed in Section 3.2, features $K$ distinct embedding layers for each quantization layer. During training, we selectively focus on a random quantizer layer $k \in [1, K]$. Token inputs are formed by embedding each token from the preceding layers $x_0^{k-1}$ and aggregating these embeddings. These inputs, combined with corresponding text embeddings and a layer indicator $k$, feed into the residual transformer $p_\phi$, which predicts the tokens of the $k$-th layer in parallel. The primary training objective is captured by:

$$\mathcal{L}_{\text{res}} = \sum_{k=1}^{K} \sum_{i=1}^{L} -\log p_\phi(x_i^k | x_i^{1:k-1}, w_s, k).$$

The parameter between the $k$-th prediction layer and the subsequent $(k + 1)$-th motion token embedding layer are shared, which simplifies the architecture and leverages feature continuity across layers.

## 4 EXPERIMENTS

We conduct extensive comparisons to evaluate the performance of our methods across various motion-relevant tasks and datasets. Detailed information on dataset configurations, evaluation metrics, and implementation nuances is available in Section 4.1. Our evaluation begins with a motion composition benchmark, where our approach is compared against existing state-of-the-art (SOTA) models across two datasets (Section 4.2). Subsequently, we focus on the text-to-motion task, contrasting our results with SOTAs that are specifically designed for single-motion generation as opposed to motion composition. Finally, we ablate some important components and techniques in our method (Section 4.3). Additional qualitative results, user studies, and extended implementation details are included in the supplementary materials.

### 4.1 EXPERIMENTAL SETUP

#### 4.1.1 DATASETS.

General motion synthesis supports a wide range of task settings, and as such, we leverage existing datasets along with a modified benchmark to comprehensively evaluate MotionStream. Our study focuses on two prominent text-to-motion datasets: HumanML3D Guo et al. (2022a) and BABEL Punnakkal et al. (2021). HumanML3D provides rich textual descriptions for each motion sequence, facilitating the direct mapping between natural language inputs and 3D human motion. In contrast, BABEL segments each motion sequence into multiple atomic components, each annotated with fine-grained textual labels, including transitions, enabling more granular control over motion generation.

| Methods | R Precision↑ | | | FID↓ | MMDist↓ | Diversity→ | MModality↑ |
|---|---|---|---|---|---|---|---|
| | Top1 | Top2 | Top3 | | | | |
| Real | $0.511^{\pm.003}$ | $0.703^{\pm.003}$ | $0.797^{\pm.002}$ | $0.002^{\pm.000}$ | $2.974^{\pm.008}$ | $9.503^{\pm.065}$ | - |
| T2M | $0.457^{\pm.002}$ | $0.639^{\pm.003}$ | $0.740^{\pm.003}$ | $1.067^{\pm.002}$ | $3.340^{\pm.008}$ | $9.188^{\pm.002}$ | $2.090^{\pm.083}$ |
| MotionDiffuse | $0.491^{\pm.001}$ | $0.681^{\pm.001}$ | $0.782^{\pm.001}$ | $0.630^{\pm.001}$ | $3.113^{\pm.001}$ | $9.410^{\pm.049}$ | $1.553^{\pm.042}$ |
| MDM | $0.320^{\pm.005}$ | $0.498^{\pm.004}$ | $0.611^{\pm.007}$ | $0.544^{\pm.044}$ | $5.566^{\pm.027}$ | $\underline{9.559}^{\pm.086}$ | $\mathbf{2.799}^{\pm.072}$ |
| MLD | $0.481^{\pm.003}$ | $0.673^{\pm.003}$ | $0.772^{\pm.002}$ | $0.473^{\pm.013}$ | $3.196^{\pm.010}$ | $9.724^{\pm.082}$ | $\underline{2.413}^{\pm.079}$ |
| MotionGPT | $0.492^{\pm.003}$ | $0.681^{\pm.003}$ | $0.778^{\pm.002}$ | $0.232^{\pm.008}$ | $3.096^{\pm.008}$ | $\mathbf{9.528}^{\pm.071}$ | $2.008^{\pm.084}$ |
| T2M-GPT | $0.491^{\pm.003}$ | $0.680^{\pm.003}$ | $0.775^{\pm.002}$ | $0.116^{\pm.004}$ | $3.118^{\pm.011}$ | $9.761^{\pm.081}$ | $1.856^{\pm.011}$ |
| ReMoDiffuse | $0.510^{\pm.002}$ | $0.698^{\pm.002}$ | $0.795^{\pm.004}$ | $0.103^{\pm.004}$ | $2.974^{\pm.016}$ | $9.018^{\pm.075}$ | $1.795^{\pm.043}$ |
| MoMask | $\underline{0.521}^{\pm.002}$ | $\mathbf{0.713}^{\pm.002}$ | $\mathbf{0.807}^{\pm.002}$ | $\mathbf{0.045}^{\pm.002}$ | $\underline{2.958}^{\pm.008}$ | $9.679^{\pm.063}$ | $1.241^{\pm.040}$ |
| MotionStream | $\mathbf{0.522}^{\pm.003}$ | $\mathbf{0.713}^{\pm.003}$ | $\underline{0.806}^{\pm.002}$ | $\underline{0.057}^{\pm.003}$ | $\mathbf{2.903}^{\pm.010}$ | $9.303^{\pm.074}$ | $1.818^{\pm.069}$ |

Table 2: Comparison of text-to-motion on HumanML3D Guo et al. (2022a). The empty MModality indicates *Real* motion is deterministic. The arrows ($\rightarrow$) indicate that closer to *Real* is desirable. **Bold** and underline indicate the best and the second best result on text-to-motion task.

For motion representation, we adopt the format outlined in Guo et al. (2022a), encompassing root velocity, joint coordinates, joint rotations, joint velocities, and foot contact information.

### 4.1.2 EVALUATION METRICS

We assessed the performance of our method using several key metrics, adhering to established evaluation protocols from prior work Guo et al. (2022a; 2023), to comprehensively evaluate motion quality, generation diversity, and text-to-motion alignment. (1) Motion Quality: We primarily utilize Frechet Inception Distance (FID), leveraging a feature extractor Guo et al. (2022a) to measure the distributional distance between the generated motions and the ground truth motions, indicating overall realism. (2) Generation Diversity: The Diversity (DIV) metric quantifies the variance across the generated motion features to assess the diversity of generated motions. In addition, MultiModality (MM) measures the diversity of generated motions corresponding to identical text descriptions, capturing the model's ability to generate multiple plausible motions under the same condition. (3) Text-Motion Alignment: To evaluate the alignment between text and motion, we employ motion-retrieval precision (R-Precision), which gauges the accuracy of matching between text prompts and motions based on Top-1/2/3 retrieval accuracy. We also measure Multi-modal Distance (MM Dist), which quantifies the distance between the embeddings of motions and their corresponding textual descriptions. In our evaluation, both the motion sequences and their textual descriptions were projected into a shared latent space using the evaluator provided by HumanML3D Guo et al. (2022a). To evaluate the quality of transitions between generated motion sequences $\hat{m}_{i-1}$ and $\hat{m}_i$, we define transitions as a sequence of consecutive poses $\{x_{L_i} - L_{tr}/2, \ldots, x_{L_i} + L_{tr}/2 - 1\}$, where $L_{tr}/2$ frames overlap with both $\hat{m}_{i-1}$ and $\hat{m}_i$. To further assess the smoothness of these transitions, we incorporate jerk—the time derivative of acceleration—following the methodology outlined in Barquero et al. (2024). Peak Jerk (PJ) captures the maximum jerk value recorded across all joints during the transition, highlighting abrupt changes in motion. Area Under the Jerk (AUJ) quantifies the cumulative deviation from natural human movement. It is computed as the sum of L1-norm differences between the instantaneous jerk of the generated motion and the average jerk observed in the dataset, offering a measure of motion smoothness throughout the transition. All metrics were averaged over 10 independent trials, with results reported alongside 95% confidence intervals to ensure statistical robustness and reliability.

### 4.1.3 IMPLEMENTATION DETAILS.

The motion tokenizer's encoder and decoder share similar architectures, both comprising 3 layers of ResNet blocks, each containing causal convolutions and skip connections. The quantizer consists of 6 residual codebook layers, each with 1,024 motion tokens of dimensionality 8, applied to both the HumanML3D and BABEL datasets. The Mask Transformer and Residual Transformer architectures comprise six layers of transformer blocks, incorporating self-attention and cross-attention mechanisms. Each attention layer utilizes 6 heads with a model dimensionality of 384. We em-

| | Subsequence | | | | Transition | | | |
|---|---|---|---|---|---|---|---|---|
| | R-prec ↑ | FID ↓ | Div → | MM-Dist ↓ | FID ↓ | Div → | PJ → | AUJ ↓ |
| GT | $0.796^{\pm0.004}$ | $0.00^{\pm0.00}$ | $9.34^{\pm0.08}$ | $2.97^{\pm0.01}$ | $0.00^{\pm0.00}$ | $9.54^{\pm0.15}$ | $0.04^{\pm0.00}$ | $0.07^{\pm0.00}$ |
| MoMask | $0.787^{\pm0.003}$ | $0.08^{\pm0.02}$ | $9.56^{\pm0.11}$ | $2.99^{\pm0.07}$ | $2.93^{\pm0.02}$ | $8.20^{\pm0.10}$ | $1.40^{\pm0.01}$ | $2.10^{\pm0.03}$ |
| MoMask w/ Interpolation | $0.756^{\pm0.005}$ | $0.14^{\pm0.04}$ | $9.42^{\pm0.12}$ | $3.15^{\pm0.01}$ | $2.92^{\pm0.09}$ | $8.15^{\pm0.09}$ | $0.05^{\pm0.00}$ | $0.95^{\pm0.02}$ |
| Ours | $\mathbf{0.719}^{\pm0.005}$ | $\mathbf{0.13}^{\pm0.02}$ | $\underline{9.27}^{\pm0.11}$ | $\mathbf{3.36}^{\pm0.01}$ | $\mathbf{2.56}^{\pm0.05}$ | $7.93^{\pm0.05}$ | $\mathbf{0.05}^{\pm0.01}$ | $0.38^{\pm0.03}$ |
| Ours w/o Code Masking | $0.615^{\pm0.005}$ | $1.56^{\pm0.04}$ | $8.79^{\pm0.08}$ | $4.13^{\pm0.02}$ | $4.59^{\pm0.10}$ | $\underline{8.22}^{\pm0.11}$ | $0.26^{\pm0.00}$ | $2.85^{\pm0.09}$ |
| Ours w/o Double Round | $0.671^{\pm0.004}$ | $0.19^{\pm0.03}$ | $\mathbf{9.33}^{\pm0.10}$ | $3.66^{\pm0.02}$ | $\underline{2.68}^{\pm0.09}$ | $7.92^{\pm0.06}$ | $0.05^{\pm0.00}$ | $\mathbf{0.33}^{\pm0.01}$ |
| Ours w/o Memory Tokens | $\underline{0.685}^{\pm0.004}$ | $\underline{0.29}^{\pm0.01}$ | $9.58^{\pm0.12}$ | $\underline{3.61}^{\pm0.01}$ | $3.45^{\pm0.10}$ | $8.29^{\pm0.09}$ | $\underline{0.20}^{\pm0.00}$ | $\underline{0.97}^{\pm0.08}$ |

Table 3: Ablation Study on the Code Masking, Double Round Training Paradigm in the Causal Motion Tokenizer and Memory Tokens Applied to the HumanML3D Dataset. ($cf$. Table 1 for notations.

| Methods | R Precision↑ | | | FID↓ | MMDist↓ | Diversity→ | MModality↑ |
|---|---|---|---|---|---|---|---|
| | Top1 | Top2 | Top3 | | | | |
| Real | $0.511^{\pm.003}$ | $0.703^{\pm.003}$ | $0.797^{\pm.002}$ | $0.002^{\pm.000}$ | $2.974^{\pm.008}$ | $9.503^{\pm.065}$ | - |
| Baseline | $0.516^{\pm.003}$ | $0.708^{\pm.003}$ | $0.803^{\pm.002}$ | $0.077^{\pm.004}$ | $2.929^{\pm.008}$ | $\mathbf{9.310}^{\pm.071}$ | $1.834^{\pm.070}$ |
| Compress $R=2$ | $\mathbf{0.522}^{\pm.003}$ | $\mathbf{0.713}^{\pm.003}$ | $\mathbf{0.806}^{\pm.002}$ | $\mathbf{0.057}^{\pm.003}$ | $\mathbf{2.903}^{\pm.010}$ | $9.303^{\pm.074}$ | $\mathbf{1.818}^{\pm.069}$ |
| Compress $R=4$ | $0.510^{\pm.003}$ | $0.701^{\pm.002}$ | $0.800^{\pm.001}$ | $0.116^{\pm.004}$ | $2.959^{\pm.007}$ | $9.259^{\pm.079}$ | $1.900^{\pm.088}$ |
| Baseline(In-context) | $0.499^{\pm.003}$ | $0.688^{\pm.003}$ | $0.785^{\pm.002}$ | $0.065^{\pm.003}$ | $3.028^{\pm.008}$ | $9.575^{\pm.065}$ | $1.170^{\pm.044}$ |
| adaLN-Zero | $0.441^{\pm.003}$ | $0.630^{\pm.002}$ | $0.731^{\pm.002}$ | $0.088^{\pm.005}$ | $3.377^{\pm.012}$ | $9.635^{\pm.082}$ | $1.104^{\pm.051}$ |
| Cross-Attention | $\mathbf{0.522}^{\pm.003}$ | $\mathbf{0.713}^{\pm.003}$ | $\mathbf{0.806}^{\pm.002}$ | $\mathbf{0.057}^{\pm.003}$ | $2.903^{\pm.010}$ | $9.303^{\pm.074}$ | $1.818^{\pm.069}$ |

Table 4: Ablation Study on the Token Compression Factor $R$ and condition injection architecture in the Mask Transformer Applied on the HumanML3D Dataset.

ploy the ViT-B/32 model for text encoding in the Mask Transformer and Residual Transformer. For training, the subsequence lengths are set to a minimum of 40 frames and a maximum of 196 frames for the HumanML3D dataset, and 40 to 200 frames for the BABEL dataset. The transition length $L_{tr}$, as defined in the evaluation, is set to 30 frames for BABEL and 60 frames for HumanML3D. In addition, all models are trained using the AdamW optimizer. The motion tokenizers are trained with a learning rate of $2 \times 10^{-4}$ and a mini-batch size of 256. Similarly, both the Mask Transformer and Residual Transformer are trained with a learning rate of $2 \times 10^{-4}$ and a mini-batch size of 256 for each training stage. The motion tokenizer is trained for 1,500 epochs, while the Mask Transformer and Residual Transformer undergo 150 and 200 epochs of training, respectively. All training processes are conducted on a cluster of 8 Tesla V100 GPUs.

## 4.2 QUANTITATIVE ANALYSIS

**Comparisons on Motion Composition.** Table 1 demonstrate the motion composition from multiple texts with the state of the art methods in HumanML3D dataset. In HumanML3D dataset, our model outperforms the other methods in subsequence quality (FID), text alignment (R-prec and MM-Dist) and transition smoothness (PJ, AUJ). In addition, as shown in Fig. 4, our method realizes both vivid subsequence and smooth transition generation between motion sequences with low generation latency even when motion sequences accumulated.

**Comparisons on Single Text-to-Motion.** The text-to-motion task focuses on generating human motion sequences from a given single text input, without requiring the composition of multiple motion sequences. We compare the performance of our proposed method against state-of-the-art (SOTA) approaches Guo et al. (2022a); Tevet et al. (2022); Xin et al. (2023); Zhang et al. (2023b); Jiang et al. (2023); Zhang et al. (2022; 2023c); Guo et al. (2023) using the HumanML3D dataset and the recommended evaluation metrics Guo et al. (2022a). Results are reported with 95% confidence intervals, computed over 20 repeated runs. The majority of the comparative results are directly sourced from the respective papers or the benchmark presented in Guo et al. (2023). Section 4.1.1 provides a detailed summary of the comparison, where our method demonstrates competitive performance across most metrics. Additionally, our approach effectively handles smooth transitions

between motion sequences when multiple text conditions are provided as input, a capability that existing SOTA methods for single text-to-motion generation lack.

### 4.3 Ablation Studies

**Motion Tokenizer.** We evaluate the effectiveness of the code masking and double round training paradigm for the motion tokenizer, as introduced in Section 3.1. As demonstrated in Table 3, the model trained on single motion reconstruction, without the double round training paradigm, fails to generate plausible motion. This is attributed to the decoder not being properly trained to differentiate between causal motion tokens across varying temporal positions. Furthermore, incorporating masking within each motion codebook significantly enhances the robustness of the motion decoder, leading to more reliable motion generation.

**Mask Transformer.** Initially, we investigate the efficacy of incorporating memory tokens within the Mask Transformer. The findings, detailed in Table 3, affirm the beneficial impact of memory tokens on model performance. Subsequently, we assess the effect of code compression on single motion generation tasks. This evaluation contrasts a baseline scenario, where all motion tokens are directly inputted into the transformer without compression, against scenarios where compression factors of $R = 2$ and $R = 4$ are applied. According to the results presented in Section 4.1.3, the compression factor $R = 2$ yields superior motion generation performance. Finally, we compare different condition injection strategies for transformers, specifically within the Mask Transformer and Residual Transformers, referencing designs from Guo et al. (2023) and Peebles & Xie (2023). The comparative results, also shown in Section 4.1.3, indicate that our architectural approach outperforms the alternatives, establishing its effectiveness in motion generation tasks.

## 5 Disscusion

In this paper, we introduced MotionStream, a novel motion-streaming framework designed to generate seamless and continuous motions that accurately reflect the semantic nuances of continuous text input. Leveraging a causal motion tokenizer based on a Residual Vector Quantized Variational Autoencoder (RVQ-VAE), we have successfully constructed a dynamic and responsive motion generation system. The dual transformer scheme implemented in MotionStream—comprising a BERT-like Masked Transformer and a Residual Transformer—enables precise prediction and synthesis of motion tokens from textual descriptions, ensuring high semantic fidelity and motion quality.

The current implementation of MotionStream is restricted to processing purely descriptive motion inputs rather than high-level instructions, which limits its applicability in end-to-end storytelling contexts. Additionally, the model's scope is confined to human body movements, excluding more diverse skeletal structures such as those of animals, as well as lacking detailed representation of facial and hand gestures. Future enhancements will focus on broadening the input capabilities to include abstract and narrative-driven instructions, thereby enriching the storytelling potential of the system. We also aim to extend the model's applicability to a wider range of biological forms by incorporating diverse skeletal models and enhancing the precision of facial and hand motion generation. These advancements will significantly expand the usability and versatility of our motion generation technology.

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

# A  APPENDIX

## A.1  TEMPORAL POSITION'S IMPACT ON CAUSAL TOKENS

We assess the impact of temporal positions on causal tokens by randomly masking motion tokens at various points throughout the sequence and evaluating the reconstruction performance. Specifically, we progressively mask 10% of the tokens, starting from the beginning to the end of the sequence.

| Position | Reconstruction | | | |
|---|---|---|---|---|
| | FID↓ | MPJPE↓ | PAMPJPE↓ | ACCL ↓ |
| Baseline | 0.01 | 23.58 | 18.46 | 7.97 |
| 0-10 | 0.10 | 62.04 | 33.82 | 11.56 |
| 10-20 | 0.05 | 49.62 | 30.23 | 11.14 |
| 40-50 | 0.06 | 47.34 | 31.09 | 11.17 |
| 70-80 | 0.03 | 42.03 | 30.24 | 11.07 |
| 90-100 | 0.02 | 35.81 | 27.07 | 10.81 |

## A.2  ABLATION ON MOTION TOKENIZER.

We ablate the motion tokenizer $\mathcal{V}$ of our models, studying the size $K$ of motion codebooks. We also compare this VQ-VAE with other VAE models in previous works Pavlakos et al. (2019); Petrovich et al. (2021b); Xin et al. (2023), as shown in Appendix A.2. This comparison demonstrates the improvement of VQ-VAE on motion reconstruction. With this ablation studies on the codebook size $K$, we thus select $K = 512$ for most experiments.

| Method | Reconstruction | | | |
|---|---|---|---|---|
| | FID↓ | MPJPE↓ | PAMPJPE↓ | ACCL ↓ |
| K=1024, d=128 | 0.147 | 48.510 | 39.504 | 10.247 |
| K=1024, d=64 | 0.018 | 34.661 | 29.621 | **7.284** |
| K=1024, d=16 | 0.009 | 33.206 | 29.029 | 7.832 |
| K=1024, d=8 | **0.005** | **33.070** | **27.470** | 7.125 |
| K=1024, d=4 | 0.012 | 43.063 | 34.162 | 7.380 |
| K=1024, d=2 | 0.015 | 51.263 | 41.381 | 9.353 |
| K=512, d=8 | 0.007 | 40.189 | 29.395 | 6.560 |
| K=1024, d=8 | 0.005 | 33.070 | 27.470 | 7.125 |
| K=2048, d=8 | 0.004 | **30.276** | 25.976 | 6.921 |
| K=4096, d=8 | **0.003** | 31.493 | 25.899 | **5.912** |
| K=8192, d=8 | 0.005 | 32.679 | **24.557** | 6.575 |

Table 5: Evaluation of our motion tokenizer on the motion part of HumanML3D Guo et al. (2022a) dataset. We follow MLD Xin et al. (2023) to evaluate our VQ-VAE model $\mathcal{V}$: MPJPE and PAMPJPE are measured in millimeter. $K$ indicates the codebook size, $d$ indicates the codebook dimension.

