# OpenReview forum: "Causal Motion Tokenizer for Streaming Motion Generation"
_ICLR.cc/2025/Conference — ICLR 2025 Conference Withdrawn Submission_

### Official Review · Reviewer_iuYe · 2024-10-26

**Soundness:** 2
**Presentation:** 2
**Contribution:** 3
**Rating:** 5
**Confidence:** 4

**Summary:**

This paper presents MotionStream, a framework for real-time motion generation that produces seamless, streaming human motions based on continuous text input. The core innovation lies in the Causal Motion Tokenizer, built upon a residual vector quantized variational autoencoder (RVQ-VAE), which enables effective long-sequence handling and smooth transitions between motion segments. MotionStream employs a dual-transformer architecture: a Masked Transformer to generate base-layer tokens and a Residual Transformer to handle additional layers, ensuring high semantic fidelity and smooth, continuous motion.

**Strengths:**

1. The proposed method demonstrates competitive performance on the HumanML datasets.

2. This framework builds a sequential motion generation method that allows for the bridging of motions between multiple textual instructions.

3. The framework diagram look beautiful.

**Weaknesses:**

1. The authors emphasise causal convolution several times in the paper (including in the title), but there is no detail on how the proposed method demonstrates causality 3between variables, nor is there a corresponding experimental analysis.

2. The ablation experiments in this paper are inadequate. Disabling Code Masking leads to significant performance degradation as shown by the results in Table 3, indicating that it is the proposed method is the key technique. Is there a separate ablation experiment analysed for mask mechanism.

3. The model seems to have been validated only on the HumanML dataset, is there more experimental data from other datasets?

4. This paper does not have state-of-the-art methods for visualising qualitative analysis.

**Questions:**

1. We found that the performance of the proposed methods in Tables 2 and 3 is close to that of MoMask. What are the advantages of the proposed method over MoMask?

2. The proposed method seems to have an advantage in terms of efficiency. It is possible to report the respective average inference times with state-of-the-art methods under the same equipment conditions.

Here's some typo:

1. Line 75. Extra commas.

2. Line 252. It seems like there should be a line break.

3. Line 435. FID 0.08 in MoMask seems to be lower.

---

### Official Review · Reviewer_8nBw · 2024-10-31

**Soundness:** 2
**Presentation:** 2
**Contribution:** 2
**Rating:** 5
**Confidence:** 3

**Summary:**

This paper presents MotionStream, a novel approach for real-time, streaming human motion generation based on text inputs. The authors address the challenge of generating continuous motion with smooth transitions, particularly in a text-driven context. The core innovation is a Causal Motion Tokenizer, built on a residual vector quantized variational autoencoder (RVQ-VAE) with causal convolution, which facilitates handling long sequences and smooth transition. Additionally, the framework employs Masked Transformer and Residual Transformer modules to enhance motion token efficiency. The model achieves real-time generation capabilities with ∼0.2s.

**Strengths:**

- Apply causal convolution to tackle the issue in current masked modeling methods where relying on future motion tokens can influence previously generated motion, especially in real-time generation.
- Add various techniques to the model, such as Code Masking, Double Round, and Memory Tokens.

**Weaknesses:**

- The main contribution seems to be only change MoMask [1] decoder to causal convolution.
- MMM [2] also has a similar architecture which has shown the "Long Sequence Generation" which can be easily applied for realtime steaming generation. Also they reported generation time 0.081s which seem to be faster than MotionStream which is 0.2s
- "Long Sequence Generation" in MMM [2] and DoubleTake in PriorMDM [3] generate the transition separately to interpolate transition between 2 motions as Figure (2.a). MotionStream, however, uses Memory Tokens that carry information from one motion chunk to the next. This could cause "motion leakage," where motion details from an earlier chunk influence a later, unrelated motion. For example, if the first prompt is "a person jumps and kicks legs" and the second is "a man crawls forward," the Memory Tokens might bring in jumping information that conflicts with the crawling motion, making the transition less smooth. Could you clarify how MotionStream handles this potential issue?
  - I am looking for the explanation of why [Fig 2.c] Streaming  ("Memory Tokens") is better than [Fig 2.a] Fixed Length. Figure 2 points it out but no explanation. And it will be the best to support the claim with ablation study and qualitative results.
- No qualitative results comparing to existing methods.

[1] Chuan Guo, Yuxuan Mu, Muhammad Gohar Javed, Sen Wang, and Li Cheng. Momask: Generative masked modeling of 3d human motions. Proceedings of the IEEE/CVF Conference on Computer Vision and Pattern Recognition
[2] Ekkasit Pinyoanuntapong, Pu Wang, Minwoo Lee, and Chen Chen. Mmm: Generative masked motion model. In Proceedings of the IEEE/CVF Conference on Computer Vision and Pattern Recognition (CVPR).
[3] Shafir, Yoni and Tevet, Guy and Kapon, Roy and Bermano, Amit Haim. Human Motion Diffusion as a Generative Prior. The Twelfth International Conference on Learning Representations.

**Questions:**

- How is the setting of "MoMask w/ Interpolation" in Table 3?
- MotionStream still requires input length, correct? If yes, I think "(2.a) Fixed Length" vs "(2.c) Streaming" is misleading.

---

### Official Review · Reviewer_xf7Q · 2024-11-04

**Soundness:** 3
**Presentation:** 3
**Contribution:** 2
**Rating:** 5
**Confidence:** 4

**Summary:**

This work focuses on the task of text-to-motion generation in a streaming way. It proposes to use a causal motion tokenizer to encode motion frames into discrete tokens. Masked transformer and residual transformer are further used to generate motions with text condition. Experiments demonstrate better generation quality and text-correspondence.

**Strengths:**

1. The paper is clear and easy to follow.
2. The proposed method to use motion masked modeling and its inpainting ability to perform streaming motion generation is straightforward and makes sense.
3. The proposed method shows superior performance compared with the SOTA method in streaming motion generation.

**Weaknesses:**

1. The novelty of this work is relatively limited. It seems largely based on the previous work MoMask [1] for the design of residual vqvae, masked transformer and residual transformer.
2. The story-to-motion is not introduced or shown in the paper.
3. The design of causal vqvae is not very clear. The transition should be mainly handled by the masked transformer. I don't see a strong ablation study to demonstrate the importance of causal convolution in vqvae, to ensure smooth transitions.

[1] Chuan Guo, Yuxuan Mu, Muhammad Gohar Javed, Sen Wang, and Li Cheng. Momask: Generative masked modeling of 3d human motions. arXiv preprint arXiv:2312.00063, 2023.

**Questions:**

1. I wonder about the insight and detailed design of causal convolution vqvae. How it could affect the performance of streaming motion generation particularly.
2. The author would better clarify their contributions other than the previous work MoMask.
3. In Table 3, it seems that the vanilla MoMask could beat the proposed method without specific adaptation. Could the author explain the reason?

---

### Official Review · Reviewer_tk78 · 2024-11-04

**Soundness:** 2
**Presentation:** 1
**Contribution:** 2
**Rating:** 3
**Confidence:** 4

**Summary:**

This work suggests a text-to-motion generation model, enabling long motion with online inputs.

It uses a combination of the following existing motion techniques:

1. Masked residual VQ-VAE (MoMask - Guo et al. - CVPR 2024, [3])
2. Autoregressive motion generation (Teach - Athanasiou et al., [1], [2], and more)

It suggests the following enhancements:
1. Enhancements to Motion Tokenizer:
   1. Code masking, double round training.
2. Enhancements to Masked transformer:
    1. Tokens compression, condition injection, memory tokens.

[1] Yang et al., Synthesizing Long-Term Human Motions with Diffusion Models via Coherent Sampling, Proceedings of the 31st ACM International Conference on Multimedia, 2023

[2] InfiniMotion: Mamba Boosts Memory in Transformer for Arbitrary Long Motion Generation, Zhang et al., 2024

[3] Pinyoanuntapong et al., Mmm: Generative masked motion model. In IEEE/CVF Conference on Computer Vision and Pattern Recognition, 2024.

**Strengths:**

- **Reproducibility:** code is attached (although implementation details and hyperparameter values are not described in the text).
- **Ablation:** thorough ablation supporting algorithmic choices.

**Weaknesses:**

- **Novelty:** This work combines existing SOTA works, and the suggested enhancements are not written clearly enough to tell whether they are innovative or incremental only.
- **Presentation:** Sloppy writing. Examples:
    - L187-196: This paragraph contains many mistakes, is repeated by Sec. 3.1, and was probably intended to be deleted
    - Figs. 1,2 - not referenced from the text and do not refer to text. This makes it hard to understand their context.
    - Unreadable too small text in Figures (particularly Fig. 4)
    - Multiple usage of an erroneous word - "steaming" (instead of streaming?)
    - L252 - missing newline before a new paragraph
    - L076- ",,"
- **References:**
    - Missing references of papers on which the essence of this work is based. See [1], [2], [3] in "Summary" above.
    - Many references are written twice (Shafir et al., Athanasiou, et al., ...)
    - Many references refer to the arxiv version rather to the venue of publication (Guo et al. 2023 --> CVPR 2024, Tevet et al. 2022 --> ICLR 2023, Shafir et al. 2023 --> ICLR 2024, ...)
- **Technical soundness:** While drawing from existing literature, several sections appear to have been adapted incompletely, resulting in gaps or Inaccuracy in crucial technical details. I will demonstrate on Sec. 3.1 only:
    - Missing info:
        - description of how the next residual value is computed (z^-z),
        - L245 - losses are mentioned but not described,
        - usage of indices (the essence of VQ discreetness) is not mentioned at all.
    - Inaccuracy: L213-214: "the high-dim **latent** vectors are downsampled": The word "latent" seems incorrect. Maybe you meant "raw"? Additionally, this sentence belongs to an earlier part of the paragraph.
- **Missing details for enhancements:** While the enhancements (listed in "Preview" above) could be considered innovations, many technical details are missing so I cannot fully assess them. Example on Sec. 3.1 only:
    - L246-252 (Code Masking): What is the masking schedule? I.e., what is the masking probability and does this probability change over time?
    - L256: What is the meaning of "resetting the causal convolution... leaving Decoder alone"?
- **Inaccurate credit:** L066 says "we first develop a causal motion tokenizer" but such a tokenizer has long been developed (see cites in L201-203).
- **Qualitative results:** many foot sliding artifacts (e.g., clip 0, time 0:17), some unnatural motions (e.g., clip1, time 2:14: getting up is unnaturally against gravity).

**Questions:**

To strengthen the paper's impact, I recommend refining the structure and presentation to enhance clarity and cohesion throughout.

---

### Official Review · Reviewer_xxuM · 2024-11-06

**Soundness:** 3
**Presentation:** 3
**Contribution:** 2
**Rating:** 5
**Confidence:** 4

**Summary:**

The paper try to addresses the challenge of continuous, text-driven human motion generation with low latency, which previous models struggled to achieve. The core contribution includes the Causal Motion Tokenizer based on RVQ-VAE and a dual transformer scheme for efficient and seamless motion generation. MotionStream outperforms existing methods in various motion quality metrics and supports applications like story-to-motion conversion.

**Strengths:**

1.Introduces a practical Causal Motion Tokenizer that improves transition smoothness for streaming generation.

2. Comprehensive empirical evaluation across datasets like HumanML3D and BABEL, showcasing superior motion composition quality and text-alignment.

**Weaknesses:**

(1) In my view, authors just extend MoMask's pipeline with some trival training strategies and use "Streaming" which is already common in related papers to tackle the problem of motion composition task; Also, it obviously doesn't provide new insights and knowledge that can benefit research community, both technical contribution and story-telling are not enough for ICLR paper.

(2) The position of figures and their layout in the paper is poor. I have to zoom in many time to see the infomation I need in Figure 2,3,4. Also, the figures and tables are far away from corresponding texts, which results in poor reading experience.

(3) In experiments section, descriptions of compared methods for motion compostion would be needed for intersted readers.

(4) Lacking qualitative comparsion and comom user study in AIGC or motion generation paper.

(5) Transitions quality is extremely important in motion composition task, but it's FID is worse than FlowMDM.

(6)  Authors claim a story-to-motion application, but I don't see any related part in the main paper.

**Questions:**

As shown in Weakness. Plus:
(1) 'Streaming' will undoubtly cause error accumulation, how does the performance of MotionStream scale with the length of the input text?
(2) In line 241, why causal structure is essential for real-time motion generation.

---

### Note · Authors · 2024-11-21

**Comment:**

Thank you for dedicating time and providing constructive feedback on our manuscript. Your insightful comments have been fully considered and highlight areas for substantial improvement, especially concerning the comparisons with existing work and the clarity of our presentation.

We have decided to withdraw our submission because our manuscript does not align with the conference's standards. This choice is made to refine our research thoroughly, informed by your invaluable feedback.

We are sincerely grateful for your comprehensive review and guidance. We hope to resubmit an improved version of our work in the future, better aligned with the conference's expectations.

**Withdrawal Confirmation:**

I have read and agree with the venue's withdrawal policy on behalf of myself and my co-authors.